

# Predicted and observed glacier pulsations in the Hissar-Alay of Central Asia

Enrico Mattea[1], Atanu Bhattacharya[2,3], Sajid Ghuffar[4], Julekha Khatun[2], Martina Barandun[1], and Martin Hoelzle[1]

[1]Department of Geosciences, University of Fribourg, Fribourg, Switzerland
[2]Department of Earth Sciences and Remote Sensing, JIS University, Kolkata, India
[3]Centre for Data Science, JIS Institute of Advanced Studies & Research, Kolkata, India
[4]Department of Space Science, Institute of Space Technology, Islamabad, Pakistan

**Correspondence:** Enrico Mattea (enrico.mattea@unifr.ch)

**Abstract.** Surge-like glacier instabilities in Central Asia remain underexplored, particularly in regions where such phenomena have low intensity or are poorly captured by existing inventories. In 1980, Glazirin and Shchetinnikov (GS1980) proposed a classification method to calculate the spatial distribution of "pulsating" glaciers in the Hissar-Alay range, predicting unstable flow at 194 candidates – over 20% of the examined sample – and claiming highly accurate detection (above 75 %). This stands in contrast to the very limited number of pulsations found in subsequent studies, which typically report fewer than 10 surge-type glaciers within the region.

Here, we address this discrepancy by reassessing the GS1980 predictions, using a newly compiled multi-sensor satellite dataset covering nearly six decades. We systematically examine glacier dynamics in the region, assessing ice flow instabilities from changes in terminus position, ice thickness, and surface morphology. We identify 171 glaciers that exhibit pulsating behavior, corresponding to 25 % of the sample. Flow instabilities tend to be modest in scale, with slow advances and long active phases (mean duration of 14 years). We find that the GS1980 model shows some ability to distinguish pulsating from stable-flowing glaciers; however, its predictive power is lower than claimed, due to the simplifying assumptions of its morphology-based approach and the uncertainties in the input data.

Our results indicate that pulsations in the region are more widespread than previously reported, but fall at the weaker end of the spectrum of glacier instability, which may not be well represented by a sharp binary classification (surge-type versus stable). As more detailed satellite records become available, we suggest that a more nuanced framework may be useful to recognize and interpret subtler instabilities of small glaciers.

## 1   Introduction

Glacier surges are loosely defined as quasi-periodic episodes of accelerated ice flow, driven by internal instability, which quickly redistribute ice from an upper reservoir to a lower receiving area (Meier and Post, 1969; Jiskoot, 2011; Herreid and Truffer, 2016). Such events alternate with longer phases of quiescent recovery, where the glacier flows slower than balance velocity and reverses the geometric changes of the active phase (Raymond, 1987). Traditional definitions of glacier surging



invoke a 10x speed-up threshold (e.g., Jiskoot, 2011), a binary classification of surge mechanisms into Alaskan (hydrology-driven) and Svalbard (thermal-driven) type (Dowdeswell et al., 1991; Murray et al., 2003; Benn et al., 2019; Ke et al., 2024), a strict distinction between surge-type and stable glaciers (e.g., Krenke, 1974; Lv et al., 2019), and an often-quoted figure of 1 % for the overall global prevalence of surge-type glaciers (Jiskoot et al., 1998). However, following the earliest descriptions of the phenomenon, subsequent observations indicate a more complex picture, including smaller speed-up magnitudes (Quincey et al., 2011, 2015; Bhambri et al., 2017; King et al., 2021), spatio-temporal patterns not fitting either Alaskan or Svalbard models (Quincey et al., 2015; Chudley and Willis, 2019; Lv et al., 2020), and more widespread prevalence of unstable flow than expected (e.g., Goerlich et al., 2020). It is generally accepted that a continuous spectrum of flow instabilities exists between the stable and surge-type end-members (Glazirin', 1978; Mayo, 1978; Herreid and Truffer, 2016; Terleth et al., 2024; Thøgersen et al., 2024). Moreover, climatic forcing and mass balance can affect surge occurrence and distribution (Hansen, 2003; Sevestre and Benn, 2015; Thøgersen et al., 2024), shape the characteristics and duration of flow instabilities (Emelyianov et al., 1974; Eisen et al., 2001; Flowers et al., 2011; Pitte et al., 2016; Mattea et al., 2025), and possibly promote surge initiation (Hewitt, 2007; Kääb et al., 2023). Overall, it is non-trivial to achieve a robust definition of glacier surges and a clear distinction from mass balance-driven advances (Herreid and Truffer, 2016; Lv et al., 2020; Thøgersen et al., 2024).

However, the indicators used to detect such glacier instabilities are quite well established: commonly mentioned are (e.g., Glazirin', 1978; Raymond, 1987; Glazovskiy, 1991; Jiskoot, 2011):

– contrasting patterns of ice thickness change between the reservoir and receiving areas

– an increase of ice velocity due to enhanced sliding, often leading to plug-like flow

– rapid terminus advance

– geomorphological signs such as distorted or looped moraines, sheared margins, increased and expanded crevassing, visible fronts and kinematic waves traveling along the glacier surface.

These indicators have been increasingly used in remote sensing studies to compile inventories of surge-type glaciers at the regional and global scale (e.g., Kotlyakov et al., 2008; Chudley and Willis, 2019; Guillet et al., 2022; Guo et al., 2023; Ke et al., 2024; Beraud et al., 2024; Guillet et al., 2025). Such inventories are often derived by analysis of a few satellite datasets (most commonly Landsat and ASTER), thanks to their multi-decadal global coverage and standardized data products. Other Earth observing satellites such as the Satellite Pour l'Observation de la Terre (SPOT) and RapidEye notably provide higher resolution, but due to their raw data format, heterogeneous spatio-temporal coverage, or other access barriers, they have so far seen more limited usage. However, they were found to significantly outperform the Landsat and ASTER products to monitor glacier dynamics, for example at the relatively small and slow Abramov glacier (Mattea et al., 2025).

Moreover, while glacier surges have been tabulated at the regional and global level via manual inspection and automated classification, comparably few studies have formulated predictive models of their spatial occurrence. Jiskoot et al. (2000) presented a logistic distribution model of surge-type glaciers in Svalbard based on geological boundaries, mass-balance conditions





and thermal regime. Sevestre and Benn (2015) examined and predicted the global prevalence of surging activity with Maxent (also logistic), based on a compilation of published surge inventories. Bouchayer et al. (2022) experimented with several machine learning models to classify surge-type glaciers in Svalbard based on geometry, climatic mass balance and runoff.

Here, we describe and evaluate an earlier attempt at predicting the spatial distribution of glacier flow instabilities in Central Asia. The study (Glazirin' and Shchetinnikov, 1980: hereafter GS1980) is virtually unknown outside Soviet glaciology; to

our knowledge, it constitutes the earliest published effort at this kind of spatial modeling. The authors compiled a dataset of "pulsating" and stable-flowing glaciers, enumerating their morphological properties according to inventory data. Then, they used these properties to train a non-parametric Bayesian classification model (Pegoev, 1977; Sect. 3.1, Appendix A). Through leave-one-out validation, they claimed highly accurate classification skill, with up to 87 and 69 % of correct predictions for stable and pulsating glaciers, respectively. Finally, they used the classifier to calculate the probability of pulsation for glaciers

in the Hissar-Alay mountain range (Sect. 2), predicting the occurrence of 194 pulsating glaciers over the study area.

Here, we conduct a systematic re-examination of glacier dynamics in the Hissar-Alay, in order to validate the 45 year old predictions of GS1980. We first inspect existing inventories of unstable glaciers in the region, both from the Soviet period and from recent studies, finding highly incomplete coverage and multiple mismatches (Sect. 2). Then, we compile a new multi-sensor satellite dataset covering the region at higher spatio-temporal resolution than provided by Landsat and ASTER products (Sect.

3). On this dataset, we investigate changes in glacier length, thickness, and geomorphology over the past 60 years, detecting the signs of glacier pulsation (Sect. 4). Based on commonly used identification criteria (Goerlich et al., 2020; Guillet et al., 2022; Guo et al., 2023), we classify glaciers as pulsating or stable flowing. We then use our observations to verify the predictions of GS1980 and assess the overall performance of that early classification model (Sect. 5). Of note, the authors adopted a broad definition of "glacier pulsation", specifically accommodating a wide spectrum of glacier instabilities – including "soft" modes

with low peak velocities and limited mass redistribution, and forgoing strict thresholds on the observed glacier changes. To reflect this, here we choose to maintain the "pulsation" terminology instead of "surge", while acknowledging the significant overlap between the two terms and the challenges of a rigorous definition.

## 2  Study region

The Hissar-Alay mountain range (Fig. 1) is located at the extreme south-western edge of Tien Shan, close to the northern

margin of the Pamirs, in the headwaters of the Syr Darya, Amu Darya and Zeravshan rivers. Shared between Uzbekistan, Tajikistan and Kyrgyzstan, the mountain range stretches west to east over about 600 km, with an average north-south extent of 80 to 150 km. It is bounded by the Ferghana valley to the north, the Alay valley to the south, the Karshi steppe to the west, and the Ferghana range to the east. Three main ridges (Turkestan, Zeravshan and Hissar) rise from the plains at the western end of the range, converging into the Alay ridge at the so-called Matcha glaciation node. The highest peaks, located

in the central area, reach an altitude of 5600 m asl. The Hissar-Alay is located in the broader arid-semiarid region of Central Asia, characterized by a generally continental climate (Barry, 1992; Aizen et al., 1995). However, in winter and spring, the Siberian anticyclonic and southwest cyclonic circulations bring relatively warm and moist air masses to the area. Therefore,



winter and annual precipitation are slightly more abundant than in the surrounding regions (Gvozdetsky and Golubchikov, 1987; Aizen et al., 1995). Precipitation increases from 350-700 mm yr$^{-1}$ in the lowlands to more than 1500-2500 mm yr$^{-1}$

above 3000 m; the given ranges correspond to a marked west-east decreasing gradient (Kotlyakov, 1977; Aizen et al., 1995). Accordingly, mean annual snow line on the western Turkestan ridge was identified in the 1970s at a height of about 3600 m asl, while in the east and south-east direction it rises to more than 4300 m asl (Rototaeva, 1979). The Hissar-Alay range features widespread, unevenly distributed glacierization (Shchetinnikov, 1981; Fig. 1). The first complete glacier inventory of the region was compiled from aerial photographs in the late 1950s, indicating a total extent of 2184 km$^2$ (Shchetinnikov,

1981). A second mapping effort in the late 1970s, based on space imagery, revealed a decrease to 1839 km$^2$ and a 15 % increase in moraine cover (Shchetinnikov, 1993a; Konovalov and Shchetinnicov, 1994). According to the Randolph Glacier Inventory version 7.0 (RGI7.0, corresponding to about year 2000; RGI 7.0 Consortium, 2023), the area encompasses 5058 glacier polygons, for a total surface of 1912 km$^2$. The apparent area increase since the 1970s does not reflect an actual glacier expansion and could be attributed to different interpretation during the compilation of the inventories, especially concerning debris-covered glaciers (also see Sect. 4.1 and Fig. 2). Total ice volume was estimated at 105 km$^3$ in 1957 and 88 km$^3$ in 1980,

using the average result of four empirical formulas (Shchetinnikov, 1993b; Konovalov and Shchetinnicov, 1994). Several individual glaciers of the Hissar-Alay were monitored *in situ* over the second half of the 20th century, including Abramov glacier with a resident program over 1967-1999 (Suslov, 1973). At present, annual glaciological observations have been re-established at several sites, and Abramov is a reference glacier with the World Glacier Monitoring Service (WGMS; Hoelzle

et al., 2017; World Glacier Monitoring Service, 2023). Those *in situ* measurements indicate a consistent pattern of negative mass balance since at least the 1960s, with almost no year of mass gain. At Abramov glacier, total mass loss exceeded 20 m w.e. between 1973 and 2023 (Barandun et al., 2015; Denzinger et al., 2021; Kronenberg et al., 2022; Mattea et al., 2025). Such a negative trend was confirmed at the regional scale by multiple studies, including geodetic assessments (Shean et al., 2020; Hugonnet et al., 2021; Fan et al., 2023), snowline-constrained modeling (Barandun et al., 2021), and geo-statistical down-

scaling of multi-source observations (Dussaillant et al., 2025). Beside mass balance, glacier dynamics in the region have long attracted attention, first with the recording of glacier terminus positions during early 20th century explorations (Mushketov, 1913; Kotlyakov, 1997; Narama, 2001, 2002), and later (since the 1960s) with the compilation of glacier inventories and the discovery of intense glacier pulsations in the nearby Pamir mountains (Dolgoushin, 1968; Dolgoushin and Osipova, 1982). As such, several early compilations of pulsating glaciers exist covering (in some cases partially) the Hissar-Alay (Suslov et al.,

1977; Dolgoushin and Osipova, 1982; Ischuk, 2018). They are based on observed active phases of pulsation, and/or inferred geomorphological evidence from ground surveys and aerial imagery. More recently, several remote sensing studies have tried to identify surge-type glaciers at the global or High Mountain Asia (HMA) scale, also including the Hissar-Alay area. These studies have applied a variety of manual and automated recognition methods on dense global datasets such as Landsat imagery, ASTER Digital Elevation Models (DEMs), ITS_LIVE ice velocity grids, or Sentinel-1 radar backscatter products (Lv et al.,

2022; Guillet et al., 2022; Guo et al., 2023; Kääb et al., 2023; Ke et al., 2024). On average, recent studies have detected fewer than five glacier pulsations in the Hissar-Alay over the past few decades (Sect. 6.4). The results usually don't match across the studies (Fig. 10); overall, the region is considered to have a low prevalence of unstable ice flow. We note that data about



pulsation of about five glaciers was integrated from the early compilations into the RGI7.0 and the Fluctuations of Glaciers (FoG) datasets, however we found several index mismatches resulting in incorrect identification (Fig. 10).

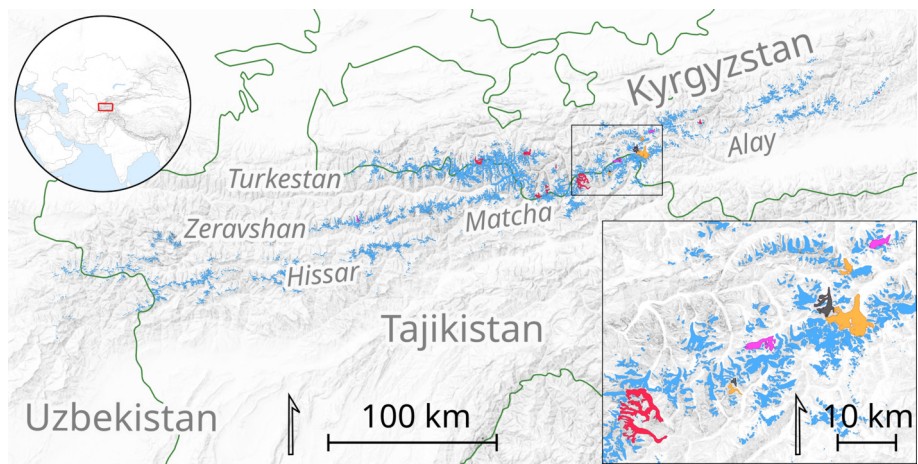

**Figure 1.** Location map of the Hissar-Alay mountain range. The inset zooms in on the region of Abramov glacier, where most previously known pulsating glaciers are located. Blue: All RGI7.0 glacier outlines in the region. Red: glaciers listed as pulsating in Soviet publications (pulsations either observed or assumed from geomorphological evidence). Pink: glaciers marked as surge-type in the RGI7.0 or by the recent remote sensing studies cited in the main text. Yellow: glaciers marked by both Soviet and modern studies. Grey: incorrectly marked surge-type glaciers in the RGI7.0 / WGMS Fluctuations of Glaciers due to mismatched indexing. Shaded relief from Copernicus DEM, produced using Copernicus WorldDEM-30 © DLR e.V. 2010-2014 and © Airbus Defence and Space GmbH 2014-2018 provided under COPERNICUS by the European Union and ESA; all rights reserved.

## 3 Data

### 3.1 List of predicted pulsations by GS1980

The GS1980 study presents the predicted distribution of pulsating glaciers in the Hissar-Alay, based on a Bayesian classification model (Pegoev, 1977). The model (detailed in Appendix A) was trained on a sample of 32 pulsating and 31 steady-flowing glaciers. Then, it was applied to calculate the probability of pulsation of all glaciers larger than 1 km$^2$ (N = 935), based on their morphological parameters. The authors tested several combinations of such morphological parameters (Glazirin', 1978), finally identifying three as the most effective to reproduce the observed training sample:

- $K$, the glacial coefficient (ratio of accumulation to ablation area extents)

- $C$, the ratio of accumulation area extent to mean width of the tongue (the latter being defined as the area between the firn line and the glacier terminus)

- $I_T$, the average surface slope of the tongue.



These parameters were derived from inventory data (Sect. 3.2), manually compiled from maps and aerial surveys according to a set of guidelines (Vinogradov et al., 1966). Based on the morphological parameters, GS1980 provide a list of 194 Hissar-Alay glaciers which are predicted to be of the pulsating type (probability > 50 % according to the classification model). Their probability of pulsation was also calculated a second time with a simpler, two-parameter version of the model, forgoing surface slope $I_T$. Additional topographic information on each glacier, such as area and length, is also reported in the list.

## 3.2 Glacier inventories

Three independent glacier inventories, systematically describing glacierization of the region, are relevant for our study:

1. The USSR glacier inventory, covering the whole area of the former Soviet Union (Grosval'd and Kotlyakov, 1969; Vinogradov, 1984; World Glacier Monitoring Service, 1989; Kotlyakov, 2019)

2. The database of A.S. Shchetinnikov, focused on the Pamir and Hissar-Alay regions (Shchetinnikov, 1981, 1997; UNDP, 2012)

3. The RGI7.0 (RGI 7.0 Consortium, 2023).

The USSR glacier inventory was published between 1965 and 1982. It consists of 108 sections within 69 books, and upon completion it was claimed to be the first comprehensive country-level inventory of glaciers (Vinogradov, 1984). It is organized in a three-level geographical subdivision (volume, issue and section), made according to drainage basins. The Hissar-Alay region is covered in issues 1 ("Syr Darya") and 3 ("Amu Darya") of volume 14 ("Central Asia"). The source data for the compilation of those issues consist of aerial photographs acquired between the 1940s and the 1960s (mostly in 1957: UNDP, 2012). Each section of the USSR inventory contains a table of individually numbered glaciers, described with their attributes: name, catchment, morphological type, and geometrical parameters such as aspect, length, surface area, estimated ice volume, minimum/maximum elevation. Also included are schematic maps with the numbered outlines of all listed glaciers, the main rivers, and the watershed lines, enabling unambiguous (albeit manual and time-consuming) localization of all glaciers.

Between 1978 and 1980, A.S. Shchetinnikov compiled a new inventory of glaciers covering both the Pamir and the Hissar-Alay. This database (hereafter referred to as Sh1980) was based on analog satellite images acquired over the same years (UNDP, 2012). It includes a set of tables with glacier indices and morphological attributes, but no figures or maps. A look-up table attempting to match the indices of Sh1980 to the USSR glacier inventory was also compiled, albeit with a large number of discrepancies and mismatches (UNDP, 2012).

Finally, the RGI7.0 is a database describing the state of global glacierization around year 2000. The Hissar-Alay are included in region 13 ("Central Asia"); there, all glacier outlines are derived from the GAMDAM glacier inventory version 2 (Sakai, 2019). In our study region, the source data are Landsat images mostly acquired in 2002. Glacier outlines were delineated manually due to the high prevalence of debris-covered glaciers. In the RGI7.0, each glacier polygon is assigned a *surge_type* index, belonging to the categories "observed", "probable", "possible", and "no evidence". The main data sources for surging information are the inventory compiled by Sevestre and Benn (2015) and the multi-factor remote sensing study of Guillet et al. (2022).



The three mentioned inventories are independent in the identification of individual glaciers and enumeration of their mor-
phological properties; correspondence between the entries is in general not one-to-one (see Sect. 4.1).

Glaciers on the GS1980 list are derived from the USSR glacier inventory, except for three sections of the latter (number 1, 2
and 5 of issue 3) which were not yet published by 1980; thus, those were replaced by those from the Sh1980 database (Glazirin'
and Shchetinnikov, 1980). The GS1980 list also provides the geometric parameters of each glacier and the corresponding values
of the predictors (Sect. 3.1) used by the classification model.

## 3.3 Optical satellite archives

In our study, we use optical satellite imagery and DEMs to look for the indicators of glacier pulsation. Such an analysis is
usually carried out on the dense, analysis-ready datasets of Landsat and ASTER. However, Mattea et al. (2025) found that
such datasets struggle to capture the dynamics of the reference Abramov glacier, which is the fifth largest of the entire Hissar-
Alay. Thus, we gathered raw optical imagery from multiple satellites with higher spatial resolution, in order to assemble a new
dataset with full spatial coverage of the study region at a repeat interval of a few years. Specifically, we acquired a total of 68
monoscopic scenes from the Satellite Pour l'Observation de la Terre (SPOT 1 to 5) (Riazanoff, 2002), distributed via the Spot
World Heritage (SWH) program (Nosavan et al., 2020) and covering the period 1986–2007, and 11 scenes from the RapidEye
constellation (Tyc et al., 2005), made available by Planet Labs' Education and Research Program (Planet Labs, 2024) and
covering 2009–2015. SPOT 5 and RapidEye provide a ground sampling distance of up to 5 m at nadir, while the earlier SPOT
1-4 achieved 10 m from 1986 to 2003. Moreover, we selected 14 image pairs from the High Resolution Stereoscopic (HRS)
instrument onboard SPOT 5, providing stereo coverage at a resolution of 5x10 m (along-track supersampling) between 2003
and 2006. To extend coverage before the digital sensor era, we included 13 declassified scans of stereo pairs from film-based
reconnaissance satellites: KH-4 CORONA Panoramic Camera (PC) and KH-9 Hexagon Mapping and Panoramic Camera (MC
and PC, respectively): Table B1), obtained from the US Geological Survey (Earth Resources Observation And Science (EROS)
Center, 2017a, b, c). Further technical details of these sensors are provided in Mattea et al. (2025), where we already processed
a subset of data from the same satellite archives in the region of Abramov glacier.

## 3.4 Analysis-ready data

To further improve spatio-temporal coverage of the study region, we integrated in our dataset several analysis-ready or previ-
ously processed satellite products. Specifically, we included orthoimages from the Sentinel-2 Multi Spectral Instrument (MSI)
at 10 m resolution (Drusch et al., 2012); we downloaded all tiles acquired over the Hissar-Alay at the end of the summer
seasons of 2018, 2021, and 2024, achieving full cloud-free coverage of the study area in those years. We also incorporated
the dataset of orthoimages and DEMs produced by Mattea et al. (2025), covering the central region of Abramov glacier (Fig.
1) at annual intervals between 1996 and 2020 from SPOT, RapidEye, Pléiades and the Indian Remote Sensing (IRS-1C/D)
program (Kasturirangan et al., 1996; Antrix Corporation Limited, 2019). In order to better estimate ice thickness changes, we
downloaded several existing global DEMs at 1 arc-second resolution: NASADEM (NASA JPL, 2020), ALOS World AW3D30
(Takaku et al., 2014; Tadono et al., 2016) and Copernicus GLO-30 (European Space Agency and Airbus, 2022). They provide





full coverage of the study region, but suffer from radar penetration in accumulation areas (NASADEM, GLO-30; Millan et al., 2015; Li et al., 2021; Bannwart et al., 2024) or from mosaicing over multi-year acquisition periods (AW3D30, GLO-30). Finally, we included nine DEMs from the High Mountain Asia DEM (along-track and cross-track pairs: HMA DEM AT / CT – Shean, 2017a, 2017b). These optical DEMs cover limited regions (encompassing only a few glaciers), but provide high spatial resolution of 8 m and well-defined acquisition times (between 2007 and 2015 in our subset).

## 4 Methods

Our study started with the identification of glaciers of interest, by combining and matching Soviet-era glacier inventories with the RGI7.0 (Sect. 4.1). Then, we processed multi-sensor raw optical satellite data (Sect. 4.2), compiling a standardized glacier-wise dataset of orthoimages and ice thickness change maps (Sect. 4.3). Subsequently, we detected the occurrence of glacier pulsations (Sect. 4.4) and calculated the performance of the GS1980 classifier (Sect. 4.5). Finally, we performed a spatio-temporal analysis of the identified pulsations (Sect. 4.6).

### 4.1 Identification of glaciers of interest

To verify the predictions of GS1980, it was necessary to identify the glaciers appearing in their published table. We decided to match the GS1980 entries with the contents of the RGI7.0, and to use the latter as the base for our analysis. The RGI is used by virtually all regional and global studies of glacier surges; its standardized, quality-controlled dataset of digital outlines is an effective starting point for further geo-processing and analysis.

The GS1980 list of pulsating glaciers is based on two different inventories (Sect. 3.1, 3.2); thus, we first matched the entries of the Sh1980 list to the USSR glacier inventory, whose maps enable localization of the glaciers of interest. We found several mismatches in the look-up table between the Sh1980 list and the USSR inventory. We corrected such indexing errors using the supplementary data found in both lists - namely, glacier names and topographical parameters, which allowed us to achieve an unambiguous, one-to-one match for all 63 Sh1980 indices appearing in the GS1980 list.

After retrieving all indices of the GS1980 list within the USSR inventory, we matched those glaciers to the polygons of the RGI7.0. It should be noted that the USSR glacier inventory was already digitized and integrated into the National Snow and Ice Data Center's Eurasian Glacier Inventory (NSIDC EGI), and subsequently into the World Glacier Inventory (Bedford and Haggerty, 1996; Haggerty et al., 1999). However, given the aforementioned indexing inaccuracies, as well as inconsistencies found during compilation of the EGI, we rather opted to unambiguously match glaciers based on their actual geometry, using the shape and overlap of each outline. To this end, we first geo-referenced the sheets of the USSR inventory encompassing the Hissar-Alay (Table 1). We geo-referenced a total of 34 maps from 8 sections, using manual ground control points (GCPs) and Thin-Plate Spline transform. Then, we inspected each of the 194 entries of GS1980, matching them to the RGI outlines based on spatial overlap (Fig. 2). Because of the different aggregation choices between the USSR inventory and RGI7.0, the relationship is not one-to-one: we found either one or several RGI7.0 polygons corresponding to each GS1980 entry, and conversely, instances of multiple USSR inventory entries corresponding to a same RGI7.0 polygon (Fig. 2). Such a many-to-





**Table 1.** Geo-referenced maps from the USSR glacier inventory.

| Volume | Issue | Section | Map numbers | Notes |
|---|---|---|---|---|
| 14 | 1 | 9 | 22, 23, 26–28, 30, 33, 35 | Northern slope of Alay ridge |
| 14 | 1 | 10 | 14–18 | Northern slope of Turkestan ridge |
| 14 | 3 | 1/2 | 10–14, 16, 18, 20 | Two sections in a single book; Zeravshan ridge |
| 14 | 3 | 4 | 14 | Southern slope of Hissar ridge: Western half |
| 14 | 3 | 5 | 10, 12 | Southern slope of Hissar ridge: Eastern half |
| 14 | 3 | 6 | 30–32, 34–37 | Southern slope of Alay ridge: Western half |
| 14 | 3 | 7 | 18, 20, 21 | Southern slope of Alay ridge: Eastern half |

many relationship would significantly complicate analysis of the GS1980 predictions. Thus, to enable further processing, we
converted the matches into a one-to-many relationship: we aggregated the spatially contiguous glacier polygons of RGI7.0 into
larger complexes, until each GS1980 entry corresponded to at most one such complex. As a result, we obtained 169 aggregated
polygons out of 226 original RGI7.0 ones. In the following, we label those 169 polygons as "predicted pulsating complexes"
(PPCs): each corresponds to one or more glaciers predicted to pulsate by GS1980. We note that this aggregation affects only a
minority of our dataset: 80 % of the 169 PPCs are composed by a single RGI7.0 polygon, and just six include more than three
polygons.

These RGI-derived PPCs constitute the basis for our inspection of recent glacier dynamics (Sect. 4.3). According to GS1980,
all other glaciers in the region larger than 1 km$^2$ are classified as stable-flowing. In order to check that prediction, we also in-
spected all such glaciers. The RGI7.0 (GAMDAM) outlines are split according to ice divides (Nuimura et al., 2015; Sakai,
2019), potentially into polygons which do not meet the 1 km$^2$ size threshold. As such, we first aggregated all spatially contigu-
245 ous RGI7.0 polygons not already identified within the PPCs, then we selected the resulting polygons larger than 1 km$^2$. We
term those "non-pulsating complexes" (NPCs). Such an aggregation also improves effectiveness of our analysis by reducing
instances of repeated inspection of a same area. Overall, between the PPCs and the NPCs, we compiled a list of 411 items to
be inspected, aggregated from 678 individual RGI polygons.

## 4.2 Processing of raw satellite archives

To inspect flow dynamics of the selected glaciers, we compiled a new dataset of orthoimages and DEMs. We processed the raw
satellite archives (Sect. 3.3) using specific pipelines adapted to each data product. Our methods to produce orthoimages out of
monoscopic SPOT and RapidEye scenes are a further development of the workflow introduced in Mattea et al. (2025). Here,
we present the main steps and the new advances that we have since implemented; additional technical details are discussed in
depth in that previous study.

For SPOT, we orthorectified the raw scenes over the NASADEM terrain using the rigorous sensor model (RSM) of Aati
et al. (2022a, b), as implemented in the open-source geoCosiCorr3D package. The method involves refinement of image



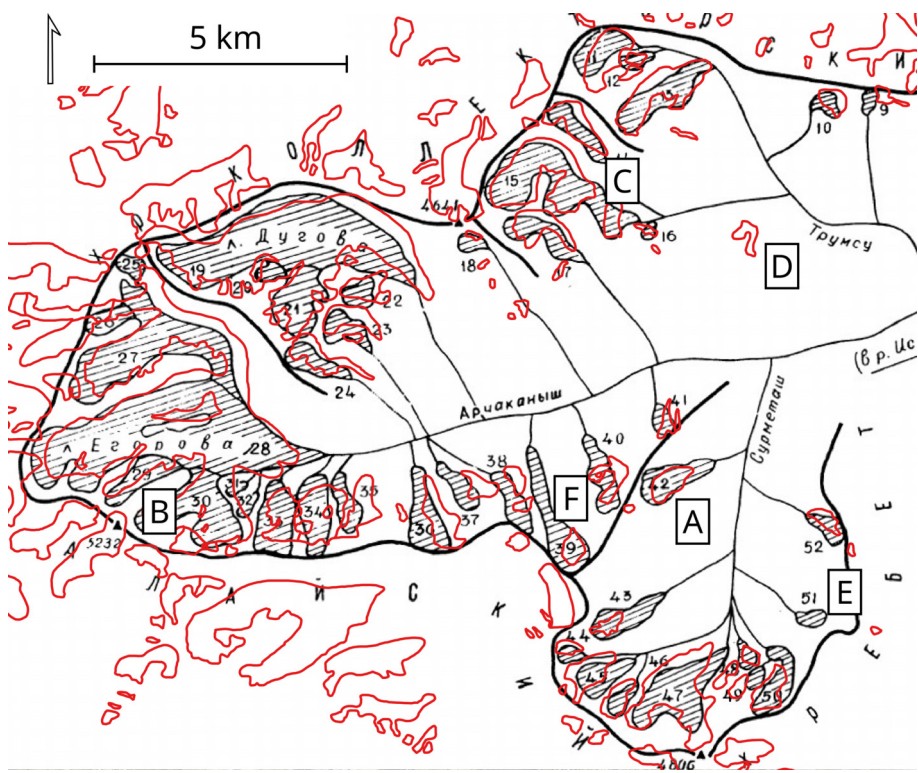

**Figure 2.** Example overlap-based matching of RGI7.0 outlines (red) to the USSR glacier inventory (background), section 14/01/09, map number 22 (northern slope of Alay ridge). A: one-to-one match. B: multiple USSR inventory entries for a single RGI7.0 polygon (Egorov glacier). C: multiple RGI7.0 polygons for a single USSR inventory entry. D: missing entry from the USSR inventory. E: missing entry from the RGI7.0. F: inconsistent classification of debris-covered terminus. Glaciers outside the catchment are excluded from the USSR inventory map.

geometry using GCPs to correct inaccuracies of the parameters reported in the scene metadata. Claimed orthorectification accuracy is up to 1/20th of pixel size (Leprince et al., 2007). Compared to the study of Mattea et al. (2025) over Abramov glacier, we did not have a high-resolution Pléiades orthoimage to use as reference over the entire region of interest. Thus, for that purpose we assembled a cloud-free Sentinel-2 mosaic from satellite passes on the 1, 6, 7 and 9 September 2018. We co-registered the individual Sentinel-2 tiles, applied a simple radiometric matching (linear least square fit over the respective overlap regions), and finally stitched the images into a mosaic covering the full region. Then, for each raw SPOT scene we first refined the footprint reported in the metadata by running interest point matching (image_align tool within the NASA Ames Stereo Pipeline (ASP; Beyer et al., 2018). This step found and corrected inaccuracies up to 100 km in the location reported for SPOT 1-4 HRV/HRVIR scenes, and up to 10 km for SPOT 5 HRG. Compared to Mattea et al. (2025), we bilinearly reprojected to Universal Transverse Mercator (UTM) the raw SPOT scenes (originally provided with rotated longitude/latitude coordinates), which significantly improved the matching step. After computing the refined scene footprint, we produced an



ad-hoc Sentinel-2 reference for tie point detection of each SPOT scene, which again improved the matching. As such, we overcame the orthorectification issues of Mattea et al. (2025) and could generate SPOT orthoimages without having to optimize

parameters of the ASP tools for each scene. As such, the updated pipeline is on average 50 times more computationally efficient than the previous version. We also found it more robust to changes in image resolution, cloud cover, and terrain surface, successfully processing SPOT 1 scenes more than 30 years older than the Sentinel-2 reference. Finally, the improved result also allowed us to shorten the loop used by geoCosiCorr3D to refine image geometry: we implemented termination of the optimization loop upon hitting both thresholds of 0.01 and 0.5 pixel for mean signed error and root mean square error,

respectively.

RapidEye scenes were orthorectified by means of the mapproject tool from NASA ASP, based on the rational polynomial coefficient (RPC) information provided within each product. We used geoCosiCorr3D to apply fine registration of each orthoimage to the Sentinel-2 reference, by subtracting the median residual displacement along the X and Y coordinates over stable terrain (defined via the RGI7.0 polygons). We found this registration method to be more robust and computationally

efficient than our previous approach (interest point matching with singular value decomposition), which would usually require fine tuning of the parameters to achieve accurate results (Mattea et al., 2025).

Stereo scenes from reconnaissance satellites were processed into orthoimages and DEMs according to the methods of Dehecq et al. (2020); Bhattacharya et al. (2021); Ghuffar et al. (2022, 2023). Technical details of those, as well as the stereo processing of SPOT 5 HRS pairs, are unchanged compared to Mattea et al. (2025).

Finally, we produced mosaics with all the orthoimages acquired on a same day by a same sensor: after fine registration with geoCosiCorr3D, we stitched them both along-track (for longer acquisitions not fitting a single scene) and cross-track (for tandem scenes acquired by the twin instruments onboard SPOT satellites). Mosaicing provides uninterrupted coverage of glaciers located along the scene borders. Overall, our dataset contains orthoimages at 68 different dates between 1968 and 2024. We applied a similar mosaicing to the dataset of DEMs, the only major difference being in the coregistration step that

we applied with xdem (xDEM contributors, 2023; Nuth and Kääb, 2011).

### 4.3   Glacier-wise standardization

In order to examine regional glacier dynamics effectively and consistently, we implemented a glacier-wise standardization procedure for our heterogeneous and multi-sensor satellite dataset. For each PPC and NPC (Sect. 4.1), we first found the orthoimages providing coverage of the glacier terminus (defined as the lowest glacierized point on the GLO-30 DEM). Then,

from all such images we extracted a standard-sized crop surrounding the glacier outline, with a 1 km margin on all sides (but constrained between 1.5 and 4 times the glacier extent, to accommodate the largest and smallest glaciers). We performed fine geoCosiCorr3D registration of all such crops to a same reference (Sentinel-2 mosaic from 2018), then bilinearly resampled them to 5 m resolution, and finally applied Contrast-Limited Adaptive Histogram Equalization (CLAHE; Pizer et al., 1987; Van Wyk De Vries and Wickert, 2021) to normalize and standardize pixel intensities. We saved each resulting image with an

overlay of the respective RGI7.0 outline(s), to provide a fixed reference towards evaluation of glacier terminus changes. The





result is a dataset of images whose coverage, nominal resolution and visual appearance are consistent and uniform for each glacier, enabling effective inspection of the dynamics.

We applied a similar standardization pipeline to the dataset of DEMs, in order to assess ice thickness changes for each PPC and NPC. We bilinearly resampled all DEMs to a same resolution of 30 m, striking a balance between sufficient detail on small glaciers and reduced noise (thanks to downsampling) on data from declassified imagery (Mattea et al., 2025). To ensure robust processing, we discarded DEM pairs having valid data coverage below 10 % of stable terrain and 50 % of glacierized area, then we co-registered all remaining pairs (Nuth and Kääb, 2011) and computed the respective DEM differences and their uncertainties (Hugonnet et al., 2022). We removed outliers from all differences with the local hypsometric filtering of McNabb et al. (2019); while the filter is designed to preserve the elevation change signal of an altitude-dependent mass balance profile, we found it to be effective also in case of the observed glacier pulsations, since they redistribute mass longitudinally down-glacier – thus generally preserving spatial coherence within relatively small (50 m) elevation bands.

## 4.4 Detection of pulsation events

We systematically inspected the standardized datasets to detect glacier pulsations. We simultaneously examined three indicators of ice flow instability (Fig. 3): (1) terminus behavior and (2) surface morphology on the orthoimages, and (3) ice thickness changes on the DEM differences. For (1), we visually identified abrupt transitions from terminus retreat to rapid advance, typically contrasting with the behavior of neighboring glaciers. For (2), we looked for actively deforming medial moraines and sudden increases in surface crevassing. We only considered present, actively observed geomorphological indicators of unstable flow, excluding relict features suggestive of previous events (such as static looped moraines or areas of downwasting dead ice). For (3), we detected the usual patterns of downstream mass redistribution, with thinning (thickening) in the reservoir (receiving) areas during the active phase, and the inverse pattern during recovery. We performed this analysis qualitatively rather than with a quantitative classification, because the actual spatial patterns are highly dependent on glacier shape, the presence of surface features (such as séracs or depressions), and surface mass balance over the time interval of the DEM difference (Mukherjee et al., 2017; Goerlich et al., 2020; Guo et al., 2023). In all the retained cases, the mass transfer signal was readily apparent, with thickness change exceeding (by a factor of two or more) the respective uncertainty. We classified glaciers as pulsating based on simultaneous evidence from at least two out of the three indicators (e.g., Guillet et al., 2022). While our processing is based on PPCs and NPCs (Sect. 4.1), we still highlighted the dynamics of each RGI7.0 component individually. During this work, we also inspected an additional 18 RGI7.0 outlines, fortuitously located within the selected areas of interest: these outlines were not part of any PPC or NPC, but showed indications of unstable flow according to the criteria outlined above. This brought the total number of examined RGI7.0 glacier polygons to 696, for a total glacierized area of 1263 km$^2$ – about two-thirds of the RGI7.0 surface in the Hissar-Alay.

## 4.5 Performance of the predictive model

We calculated performance of the GS1980 predictive model by comparing the counts of correct and incorrect classifications (true/false positives and negatives: TP/FP/TN/FN). Such counts are not trivial to define, because of (1) the one-to-many rela-





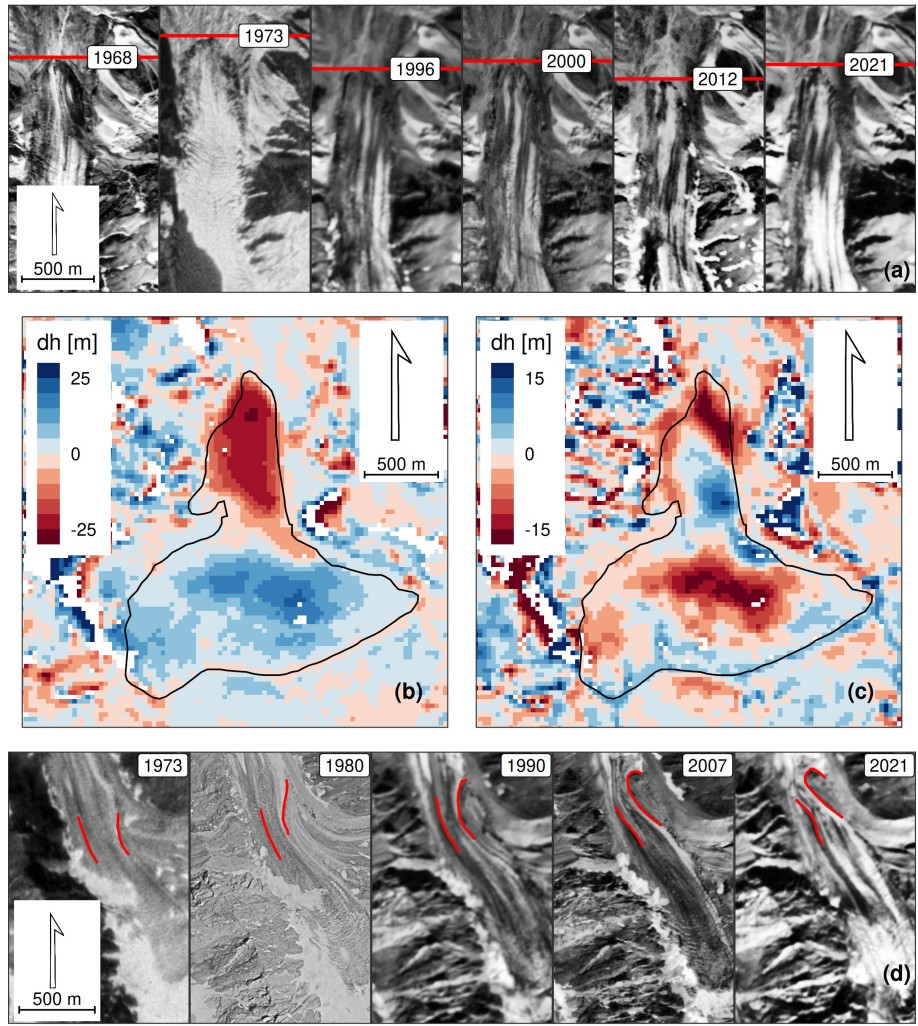

**Figure 3. (a)** Cropped subset of the standardized orthoimages dataset for glacier 02585, showing three terminus pulsations between 1968 and 2021. The horizontal red line marks the glacier terminus. Image sources: KH-4 PC, KH-9 MC, SPOT 3, IRS-1C, RapidEye, Sentinel-2. **(b)**, **(c)** Thickness change maps for glacier 02670 (black outline) between 20/08/1980 (KH-9 MC) and 06/12/2005 (SPOT 5 HRS) and between 06/12/2005 and 08/09/2020 (Pléiades), showing the mass redistribution patterns typical of a quiescent phase of recovery and of an active phase of pulsation, respectively. **(d)** Cropped subset of the standardized orthoimages dataset for glacier 01913, showing active moraine deformation indicative of unstable ice flow. Image sources: KH-9 MC, KH-9 PC, SPOT 2, SPOT 5, Sentinel 2. The three glaciers are not reported as pulsating in any previous source (see Fig. 10). We classified them as such based on at least two criteria (see main text), that is, at least a second one in addition to the one presented in the figure. All indices were shortened by removing the full prefix "RGI2000-v7.0-G-13-".

tionship between the USSR glacier inventory and the RGI, (2) the occurrence of pulsations on only some areas or branches of
335 a glacier, and (3) the choice by GS1980 to report only glaciers predicted to pulsate, which somewhat obfuscates identification



of predicted stable glaciers. As such, we tested two different methods of counting the classified glaciers, reflecting glacier aggregation according to (1) the RGI7.0, and (2) the USSR inventory, respectively. In version (1), we count as TP an RGI7.0 polygon which respects the following: we found it to be pulsating, and it belongs to one of the PPCs (Sect. 4.1). An RGI7.0 polygon is counted as FP if it belongs to a PPC but we did not find it to pulsate. By contrast, in version (2), an RGI7.0 polygon

is counted as TP even if we did not specifically observed it to pulsate, as long as we found pulsation of at least one other polygong belonging to the same PPC. Under this definition, a non-pulsating RGI7.0 polygon is an FP only if no glacier within the PPC is pulsating. Definition (1) always yields a lower count of TPs than version (2), thus a lower calculated performance of the predictive model. With both counting methods, FN and TN are defined in the same way: an FN is an RGI polygon which we found to pulsate while it is not included in any PPC, while TN is applied to all RGI7.0 polygons (of the 696 examined)

not falling into another category. We note that the majority of PPCs are actually composed of a single RGI polygon, thus the difference between the two counting methods is relatively minor (Sect. 5.3).

We applied these two counting methods to both versions of the GS1980 predictions – with two and with three predictors (Sect. 3.1). Thus, we obtained four independent counts of TP/FP/TN/FN. We measured performance of these results with standard metrics commonly used for classifiers (Saito and Rehmsmeier, 2015), including recall, specificity, and standard accuracy,

as well as metrics specifically accounting for class imbalance in the dataset: balanced accuracy, precision, and F1 score (Appendix C). The latter metrics are especially meaningful in our case: even though the GS1980 study performed leave-one-out validation on the almost balanced training dataset (Sect. 1), it later found a significant difference in prevalence between pulsating and stable glaciers in the region (194 and 741 out of 935, respectively). Formulas for all computed metrics are given in Appendix C. Finally, we also compared performance of GS1980 with that of uninformed random guessing (assuming equiprob-

able classes) as well as stratified random guessing (assuming rates of pulsation prevalence as found by GS1980 and by our study).

### 4.6 Spatio-temporal analysis

We examined the spatio-temporal distribution of the identified glacier pulsations, to compare with the GS1980 predictions as well as with results from other studies in the region and in the vicinity. We defined the temporal bounds of single pulsation

events by selecting the periods corresponding to positive identification criteria for unstable flow (Sect. 4.4).

As such, in several cases the start/termination of a pulsation are coincident with the first/last evidence of terminus advance. However, to accommodate multi-phase pulsations and seasonal dynamics (due to the inconsistent orthoimage dates), we relaxed the requirement of continuous terminus advance over individual years: we considered as a single pulsation all instances of interrupted advance resuming within a year.

Despite the use of multiple satellite sources, our dataset still has limited temporal coverage, especially in the earliest period (until 1985). This results in multi-year intervals between consecutive observations, complicating the accurate definition of start and end dates of pulsations. Thus, we calculated three estimates for the duration of each pulsation – low, mean, and high (Fig. 7). Given a pulsation observed over a few consecutive intervals, the "mean" estimate of duration assumes that the pulsation starts on the same date as the satellite pass at the start of the first interval, and terminates exactly at the satellite pass marking the





end of the last interval. The "low" estimate assumes a pulsation starts halfway through the first interval of observed instability, and ends halfway through the last interval. Finally, the "high" estimate assumes a pulsation starts halfway through the interval preceding the first unstable one, and similarly extends halfway into the next interval after the last observed signs of unstable flow.

## 5 Results

### 5.1 Overview

We found evidence of pulsating behavior at 171 of the 696 examined RGI7.0 glaciers, corresponding to a prevalence of 25 %. For the majority of glaciers, we could confirm a single pulsation event within their observed time span; however, we also detected two pulsations at 31 glaciers, and three pulsations at three glaciers, for a total of 196 recorded pulsation events. Of those, 128 (65 %) are fully contained within our dataset, that is, not already (or still) ongoing at the start (or end) of the covered 380 period. Moreover, 145 events (74 %) last longer than a single observation interval (Sect. 4.6); on average, each pulsation can be tracked over three intervals. Finally, 103 pulsations are both fully contained and longer than one interval, enabling robust calculation of the event duration.

### 5.2 Pulsations distribution and characteristics

The observed glacier pulsations are distributed very unevenly across the study region (Fig. 4). The highest prevalence is found 385 between the central Matcha region and the Abramov glacier, and a strong decrease can be seen towards the western and eastern margins of the Hissar-Alay.

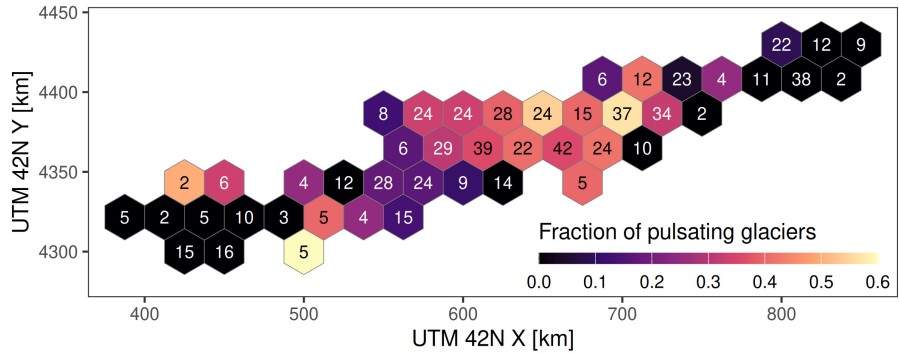

**Figure 4.** Prevalence of pulsating glaciers relative to the examined sample. Total sample size is inscribed in each tile.

The largest pulsating glacier is Archa-Bashy (27 km$^2$), while the smallest is a 0.17 km$^2$ unnamed glacier in the Turkestan ridge. Median extent of pulsating glaciers is 1.56 km$^2$, and 23 % are smaller than 1 km$^2$. Total extent of the pulsating glaciers





is 464 km$^2$, or 37 % of the surface of all examined glaciers (Sect. 4.4). The largest 26 pulsating glaciers account for half of

390 this extent. There is a significant positive correlation (Spearman's rho = 0.92) between pulsation prevalence and glacier area, with the former increasing from 1 % to 54 % between the smallest and largest area classes (< 0.25 km$^2$ and > 5 km$^2$: Fig. 5). However, the most extensive glacier complex of the study region (Zeravshan, with 93 km$^2$) is apparently stable in all its branches, as are the largest valley glaciers in the vicinity (Rama and Farakhnau). Among the pulsating glaciers, 159 exhibit terminus advance, while 12 only have internal pulsation with a stable or retreating terminus.

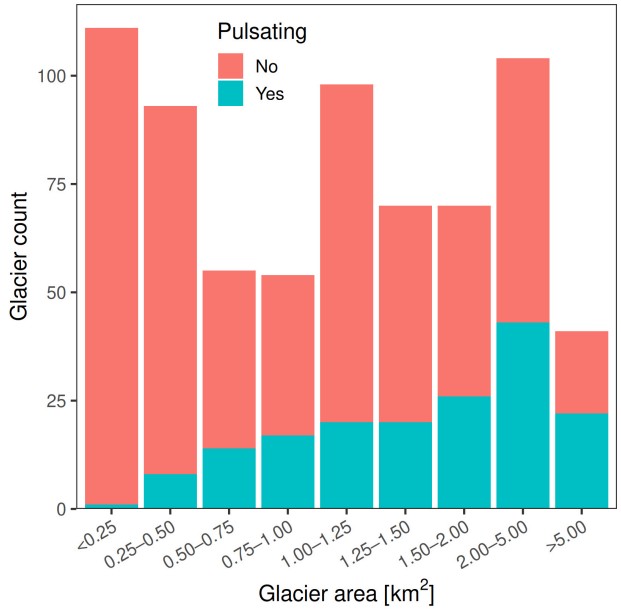

**Figure 5.** Counts of pulsating glaciers and total examined ones, by area class of the respective RGI7.0 polygon.

Over the almost 60 years of observations, two periods clearly stand out with an increased occurrence of pulsations: around year 1970 and 2004 (Fig. 6). During those years, as many as 17 and 11 % of observed glaciers are simultaneously pulsating. However, the sample of examined glaciers is relatively small in the earlier period, with coverage available for only one in three glaciers. As such, the largest number of simultaneously pulsating glaciers occurs in 2004, with a total of 79. The occurrence of pulsations is strongly reduced in the years around 1982 and towards the present, with only about 2.4 and 4.3 % of glaciers

showing evidence of pulsation, respectively. Mean duration of the active phase is 14 years over 103 pulsations with good observations (Sect. 5.1, with a standard deviation of 7.5 years (Fig. 7a). Low and high estimates (Sect. 4.6) are 9.0 and 18.4 years, respectively. For glaciers with multiple pulsations, we observed a mean interval between terminations of 31.7 years (Fig. 7b) and a mean duration of 20.1 years for the quiescent phase. Low and high estimates differ by up to 11 years due to limited temporal coverage over the early period.





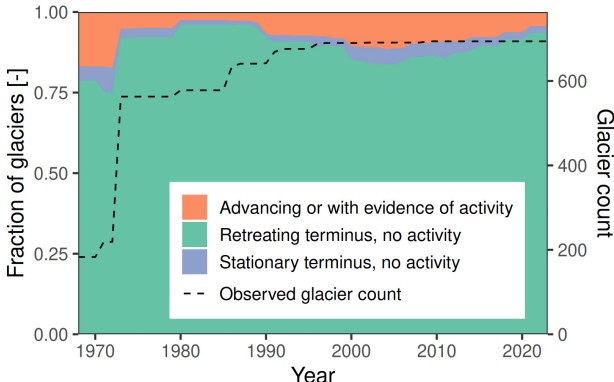

**Figure 6.** Distribution of observed glacier behavior over time. "Activity" refers to any of the indicators of active pulsation as described in Sect. 4.4 – namely, enhanced downstream mass redistribution (visible on DEM differences) or active geomorphological evidence of unsteady flow.

## 5.3  Assessment of the GS1980 predictive model

By all computed performance metrics, all versions of the GS1980 predictions are less accurate than claimed by the authors via leave-one-out validation (Fig. 8). The discrepancy is especially visible in the metrics that account for class imbalance in the dataset, such as precision and F1 score. However, by the same metrics, the GS1980 model performs significantly better than all versions of random guessing, demonstrating concrete predictive power at discriminating pulsating and stable glaciers. As expected, model performance increases both with the addition of a third predictor (mean slope of the tongue) and when using the more permissive counting of glacier pulsations (based on the USSR inventory rather than RGI7.0: Sect. 4.4).

Overall, the observed prevalence of pulsating glaciers (171 / 696 = 24.6 %) is similar to the predicted one (194 / 935 = 20.7 %). There is no clear pattern in the spatial distribution of model performance across the study region (Fig. 9a). However, the model predictions appear to become increasingly accurate with increasing glacier area (Fig. 9b), exceeding what would be expected from the correlation between area and pulsation prevalence (Fig. 5). The GS1980 model correctly anticipated the high concentration of pulsating glaciers in the central region of the domain, even though the largest glacier complexes (around Zeravshan glacier) are classified as false positives. Finally, the correlation between predicted probability of pulsation and observed prevalence (Fig. 9c) is quite weak. Spearman's rho metric is just 0.15, however it is heavily affected by the absence of observed pulsation at the two glaciers with the highest computed probability (> 95 %: rightmost class in the figure); excluding those, the same measure of correlation would rise to 0.58.




## 6 Discussion

### 6.1 Evaluation of GS1980

Our findings indicate that the GS1980 predictive model has lower accuracy than claimed by the authors, but still some distinct predictive power compared to random guessing (Fig. 8). The method is simple and computationally inexpensive (Appendix A); however, we note some serious shortcomings within its application to pulsating glaciers. Specifically, the morphological parameters underpinning the calculation can be highly uncertain and somewhat subjective, depending on the determination of accumulation area, firn line and glacier terminus from aerial imagery (Sect. 3.1). Measurement uncertainty and subjective interpretation arise in the manual compilation of inventories from just one or few optical acquisitions (Glazirin' and Shchetinnikov, 1980), which may not reflect long-term steady-state conditions over the glaciers. Moreover, the combination of data from different inventories, with heterogeneous reference times (Sect. 3.2), can introduce a mismatch both in the training dataset and in the parameters of the examined glaciers, producing inconsistent results across the study region. Even postulating a rigorous definition of the predictors, additional ambiguities arise with the identification of debris-covered glaciers and the dynamical separation of ice masses at ice divides (Fig. 2), again potentially affecting the computed values of the morphological parameters. The problem of dynamical separation appears to have been at least contemplated by the GS1980 authors, who tested the method at the dendritic Egorov glacier (Fig. 2) both on individual tributaries and on the single, contiguous complex. While the authors do not discuss the result, the computed probabilities are indeed quite sensitive to the partition of tributaries, ranging between 52 and 80 % (compared to 75 % for the whole glacier complex). Such a high sensitivity is also confirmed by our analysis of the GS1980 table (Fig. A1d). These weaknesses underscore a key shortcoming of the method, namely the assumption that pulsating behavior is mostly an expression of glacier morphology. This notion was supported by early conceptual models of glacier surging (e.g., Kamb, 1987; Fowler, 1987; Clarke, 1991). However, subsequent theoretical developments (Sevestre and Benn, 2015; Benn et al., 2019) rather attribute surge-like instabilities to a combination of climatic, topographic and geologic factors, leading to imbalances of glacier enthalpy and thus cyclic instabilities on diverse time scales (Terleth et al., 2024). While the GS1980 predictors do have an indirect link to climate (via the relative location of the firn line on the glacier), they are too crude to meaningfully represent its relationship with glacier instabilities. Overall, a lesser importance of glacier geometry is consistent with the lower predictive power found for the GS1980 model. Concretely, it is not straightforward to establish a direct link between the model predictors and modern theories of flow instability (Benn et al., 2019; Terleth et al., 2024; Thøgersen et al., 2024); going beyond our study, an analysis of the propensity to enthalpy imbalance for a realistic glacier morphology could yield some insights in that sense.

A relatively clear change could be expected for the GS1980 predictor parameters in response to recent accelerated climatic warming, with a marked reduction in accumulation area extent. Such a change would generally lead to a decrease in the computed probabilities (Fig. A1d), resulting in a smaller prevalence of predicted glacier pulsations. Continued observations of glacier dynamics in the region would be needed to support this hypothesis.



## 6.2  Pulsating glacier dynamics

Overall, the observed pulsations appear to confirm the "quiet" or "soft" nature of ice flow instabilities in the region (Glazirin',
1978; Mayo, 1978; Glazirin' and Shchetinnikov, 1980; Mattea et al., 2025); this is characterized by relatively slow and minor
terminus advances (tens to few hundreds of meters over several years, even for the largest glaciers), and usually lacks separation
of the receiving area into a static, down-wasting mass of dead ice.

The relatively widespread occurrence of glacier pulsations is consistent with the findings of Goerlich et al. (2020) over
the northern area of the Pamirs. Moreover, that study reported a peak in surging activity taking place between 2000 and 2008,
which also matches our results over the Hissar-Alay (Fig. 6). Other similarities include the observation of pulsations on glaciers
smaller than 0.5 km$^2$, and a mean duration of the active phase slightly exceeding one decade. However, the surge-type glaciers
described by Goerlich et al. (2020) are in general much larger (mean area > 10 km$^2$) and have more intense events compared
to the pulsations observed in the Hissar-Alay.

The variable number of pulsating glacier over the years, peaking around 1970 and 2004 (Fig. 6), could express the influence
of a climatic driver over the glacier dynamics. For instance, occurrence of enhanced melting – potentially triggering widespread
frictional instabilities at the mountain range scale – was also discussed by Hewitt (2007); Kääb et al. (2023); Thøgersen et al.
(2024). However, a deeper analysis of the climatic history of the region would be needed to support this consideration and rule
out the possible impact of sampling bias on this result (Goerlich et al., 2020).

A high prevalence of glacier pulsations has implications for mass balance studies in the region. Specifically, geodetic cal-
ibration or validation of regional-scale modeling studies is delicate and requires frequent updates to the used outlines, to
accommodate evolving glacier geometries (Barandun et al., 2021; Barandun and Pohl, 2023; Kronenberg et al., 2022). More-
over, Mattea et al. (2025) previously confirmed that the Abramov glacier (reference glacier in the Pamir-Alay for the WGMS)
is pulsating, casting doubt on its regional representativity. Our study expands that finding with the observation of widespread
pulsation across the region, further complicating interpretation of mass balance trends at the single glacier and regional scale.
While *in situ* measurements remain invaluable for calibration and validation of mass balance studies, information on the status
of the pulsation cycle should be researched and included for a meaningful interpretation of their results.

## 6.3  Study uncertainties and limitations

A key challenge in remote sensing studies of glacier pulsation is the correct identification of surging activity. The issue is
particularly critical in our case, since the study region is characterized by glaciers at the lower end of the spectrum of instability,
closer in appearance to stable flow or to mass balance driven advances than most other surge-type glaciers. As such, individual
mis-classifications are possible; our results (including the accuracy evaluation of GS1980) could be affected by a certain degree
of interpretation, only partially mitigated by the combined use of multiple classification criteria. This characteristic is shared by
all studies compiling inventories of glacier instabilities, which routinely find discrepancies in the evaluation of the behavior of
single glaciers (e.g., Lv et al., 2019; Goerlich et al., 2020). However, such divergences are usually minor and do not significantly
alter the conclusions of statistical and distribution analyses.



Coverage of the study region within our dataset is incomplete at the start of the period (Fig. 6, Table B1). As such, it is possible that we did not detect some pulsations taking place between the late 1960s and the late 1980s. In fact, some pulsating glaciers may be missed altogether due to their long recurrence interval, potentially exceeding the full duration of our dataset (Fig. 7b). Because annual imagery is not available for most glaciers (Sect. 4.6), the reported durations and intervals are also spread out as reported in Sect. 5.2. Moreover, the exact timing of a pulsation is subject to interpretation, whenever the active phase starts within the glacier body and then leads to terminus advance: this could lead to discrepancies with studies analyzing pulsations from flow velocity (Goerlich et al., 2020). To this respect, we did not attempt measurement of ice surface velocity from the orthoimages, because the small extent of most glaciers and the multi-annual intervals in the dataset hamper the successful application of window-based correlators (Mattea et al., 2025). However, it should be noted that changes in ice velocity are indirectly reflected in all three indicators of flow instability that we considered (Sect. 4.4). Indeed, glacier acceleration can be qualitatively inferred both in a sharp change from terminus retreat to advance, and (via the continuity equation) in the patterns of ice thickness change (Fig. 3c). Changes in flow direction and in relative velocity of contiguous ice masses drive the development of many surge-like geomorphological features such as looped and deformed moraines (Fig. 3d).

Additional uncertainties in our study arise from the processing of remote sensing data archives, especially in the case of declassified reconnaissance data affected by various types of artifacts (e.g., Dehecq et al., 2020). A systematic uncertainty analysis was performed by Mattea et al. (2025), where the accuracy and suitability of both orthoimages and DEMs were established against *in situ* measurements and independent remote sensing observations during two full pulsation cycles of Abramov glacier. Given the similar data sources and methods used here, we expect those conclusions to also hold at the mountain range scale.

One last source of uncertainty with glacier identification is given by the recurrent confusion in the indices of pulsating glaciers which are reported or compiled by previous studies. We discovered several mismatches between the Sh1980 and USSR inventories (Sect. 4.1, and also in the surge events compiled into the RGI7.0 (Sect. 2, Fig. 10). We found and corrected such mismatches via manual examination of the individual glaciers: we suggest that studies comparing their results to previous compilations should carefully check glacier identification in order to rule out potential errors of this kind.

## 6.4 Significance and comparison to previous studies

Our study has found 171 pulsating glaciers in the Hissar-Alay. After correction of indexing mismatches, only 15 were already known from past studies (including those assumed from indirect evidence), with major discrepancies in the identification of individual glaciers (Fig. 10). In particular, the HMA-wide remote sensing analyses of Guillet et al. (2022) and Guo et al. (2023) found just two and five glacier surges over 2000–2018 and 1970–2020, respectively. Rigorous inter-comparisons of surge-type glacier inventories are challenging due to the inconsistent data, methods and criteria employed by different studies (e.g., Guillet et al., 2022). Still, the large number of newly identified pulsations could be explained by several factors:

- Our selection of multi-source satellite data provides finer spatial resolution and longer temporal coverage than previous studies, enabling better detection of the low-intensity events which characterize the region.





- Larger-area studies usually set thresholds for automated detection of glacier surges, and those are probably not matched by the observed low-intensity pulsations.

- Manual identification of the highest-quality satellite scenes, later processed via a specifically-tuned, glacier-wise pipeline (Sect. 4.2), can likely extract and preserve detail better than the standardized data aggregation and automated selection methods used within HMA-wide or global studies.

The observation of pulsations at several very small glaciers ($< 1$ km$^2$; Fig. 5) supports the assumption that existing inventories of surge-type glaciers can be biased towards larger glaciers, especially due to preferential sampling of the latter (e.g., Goerlich et al., 2020). Such a bias could potentially persist into the published distribution analyses of morphological parameters (such as slope) for surge-type versus stable glaciers – analyses which are frequently used to pinpoint characteristics conducive to surging activity (e.g., Guillet et al., 2022; Guo et al., 2023). This bias at the lower end of the spectrum of instability may not be sufficient to overturn the conclusions of such analyses. However, the relevance of a sharp classification (surge-type versus stable) could be called into question for such small, slightly unstable glaciers. As higher-resolution data keeps revealing more numerous and less intense pulsations on smaller glaciers, it may become necessary to adapt such distribution analyses, for example (1) restricting the sample to a given detection threshold for glacier area and magnitude of instability, or (2) explicitly incorporating into the analysis a quantitative metric of flow instability, such as relative magnitude of speed-up and mass redistribution.

The methods we have implemented to compile the multi-source remote sensing dataset are mostly open-source and automated, enabling efficient up-scaling and potentially the detection of small glacier pulsations in other regions. In particular, cloud-free satellite mosaics and global DEMs are readily available to serve as reference for the orthorectification and stereo processing of raw scenes. The most time-consuming parts of our workflow are (1) the selection of good-quality initial raw scenes, often lacking usable cloud or snow masks, and (2) the manual inspection of orthoimages and DEM differences to assess the occurrence of pulsations. For (1), several methods are being developed to calculate quality masks even on monoscopic, panchromatic scenes (e.g., Fisher, 2014). Full automation of our multi-criterion inspection is more challenging: existing studies have used algorithmic evaluation of ice thickness changes and ice surface velocities (e.g., Guo et al., 2023; Guillet et al., 2022; Beraud et al., 2024; Guillet et al., 2025), but effective machine recognition of terminus advances and especially of surface geomorphology is still in the nascent phase (Maslov et al., 2025; Barnes et al., 2024).

## 7 Conclusions

This study presents a comprehensive assessment of glacier dynamics in the Hissar-Alay mountain range, towards verification of the predictions of glacier pulsation made by Glazirin' and Shchetinnikov (1980). By compiling a high-resolution, multi-sensor satellite dataset spanning nearly six decades, we systematically re-examined the dynamics of 696 glaciers and found evidence of pulsation in 171 of them, revealing a prevalence of 25% – a significantly higher figure than previously reported for the region.





Our analysis confirms that glacier pulsations in the Hissar-Alay are widespread but typically subdued in nature, with modest terminus advances and relatively long active phases (mean duration of 14 years). These instabilities affect glaciers of all sizes, including many smaller than 1 km$^2$, suggesting that prior inventories may have underrepresented the full extent of pulsation activity due to limitations in spatial resolution, temporal coverage, or detection criteria. An increased occurrence of pulsations in the early 2000s matches the results of previous studies in the neighboring Pamir region.

The GS1980 classification model, while less accurate than originally claimed, exhibits meaningful predictive ability in distinguishing pulsating from stable-flowing glaciers. However, its reliance on geometric parameters – which are only indirectly related to local climate – introduces important limitations, as the predictor values are subject to uncertainty and interpretation. While the model yields some insight into the distribution of glacier instabilities, it cannot fully account for the occurrence of pulsations as these are influenced by factors beyond morphology.

Finally, the frequent detection of low-intensity pulsations confirms the challenges of a sharp binary classification between stable and pulsating behavior, particularly in small glaciers. Many of the observed events belong to the lower-intensity end of a continuous spectrum of glacier flow instabilities. This suggests the need for more flexible classification frameworks that can reflect the full range of dynamical behaviors, now observable thanks to increasingly detailed satellite records.

*Code and data availability.* Raw declassified scenes can be accessed at the EarthExplorer (https://earthexplorer.usgs.gov/). DEMs and or-
565 thoimages from KH-9 MC, SPOT, and Pléiades will be made available upon publication subject to the respective access licenses. IRS and RapidEye scenes should be requested respectively from the Antrix Corporation and Planet Labs. Software pipelines for scene orthorectification, DEM generation and differencing, and post-processing analyses, will be made available in a public-access repository upon publication. Most Russian-language publications cited within the study can be accessed through https://sites.google.com/view/glaciobiblio/.

## Appendix A: The classification method of GS1980

The list of predicted glacier pulsations by Glazirin' and Shchetinnikov (1980) was computed using the non-parametric classification method of Pegoev (1977). While both publications were until now unknown outside the Soviet sphere, the method bears strong resemblance to the concepts formulated by Fix and Hodges (1952) and Parzen (1962), belonging to the family of Bayesian classifiers with Kernel Density Estimation.

In Pegoev (1977), each object in the training dataset is associated with a feature vector of predictor values, such as morpho-
575 logical parameters of a glacier, and also belongs to one of a set of possible classes, such as pulsating or non-pulsating. In order to classify a candidate object of unknown class, the method calculates a similarity score between its feature vector and that of each object in the training dataset. This set of scores is then used to formulate a soft (probability-based) prediction on the adequacy of each class to contain the candidate object.

In particular, the method calculates scores with multi-variate pseudo-Gaussian distributions centered at the feature values of
580 each training object, accommodating potential knowledge about uncertainties within the features: in log-probability form (for numerical stability: Bishop, 2006) this is expressed as



$$\xi_i = -\sum_{j=1}^{n} \left[ \frac{(x_{j0} - x_{ji})^2}{\sigma_{j0}^2 + \sigma_{ji}^2} + \frac{1}{2} \ln(\sigma_{j0}^2 + \sigma_{ji}^2) \right] \qquad (A1)$$

where index $i$ refers to the $i$-th training object, index $j$ iterates over the feature vector of length $n$, $x$ refers to the value of a feature in the vector, $x_{j0}$ is the value of feature $j$ for the candidate object, $x_{ji}$ is the corresponding value for the $i$-th training object, and $\sigma$ are the respective standard deviations.

The probability density $\xi_i$ is then inserted into Bayes' theorem in order to accommodate potential prior knowledge about the similarity with training objects:

$$P_i = q_i \cdot P_i(x_{j0}) = q_i \cdot \exp\left(\xi_i - \xi_{\max}\right) \qquad (A2)$$

where $P_i$ is the computed (posterior) similarity score, $q_i$ is the assumed prior, $P_i(x_{j0})$ is the pseudo-Gaussian probability distribution of the training object evaluated at the feature values of the candidate object, and $\xi_{\max}$ is a normalization factor, again for numerical stability.

Finally, classification probability is estimated as a linear combination (soft class membership):

$$g_l = \frac{\sum_{i=1}^{m} P_i \cdot c_{li}}{\sum_{i=1}^{m} P_i} \qquad (A3)$$

where $g_l$ is the probability of the candidate object of belonging to class $l$, index $i$ iterates over the $m$ training objects, and $c_{li}$ is 1 when training object $i$ belongs to class $l$, 0 otherwise. The class with the highest probability is then applied to the candidate object.

In the GS1980 application, there are only two classes (pulsating and not pulsating glaciers); the method is applied twice, first with three-parameter feature vectors and then with a simplified, two-parameter version (Sect. 3.1). The authors compiled a list of 32 pulsating and 31 stable glaciers to serve as training objects; to select them, they used inventories of observed pulsations and a new analysis of high-resolution aerial photographs.

Unfortunately, the list of training glaciers is omitted from Glazirin' and Shchetinnikov (1980). Moreover, the authors provide no details on both the estimated uncertainties in the feature vectors (the $\sigma_j$ in Eq. A1) and the prior probabilities (the $q_i$ in Eq. A2). Thus, full replication of the method is challenging. However, we could still analyze the probabilities and morphological parameter values reported in GS1980, in order to reconstruct the main characteristics of pulsating glaciers according to the method (Fig. A1). Distributions of the morphological parameters (Fig. A1a/b/c) do not reveal any specific patterns. The glacial coefficient $K$ appears to follow a symmetric, quasi-Gaussian distribution, unlike the other two predictors. In the two-parameter model (Fig. A1d), there is a clear maximum of computed probability for high values of $K$ (above 1.5) and intermediate values of $C$ (between 2 and 4 km), where the computed probability of pulsation is close to 100 % – meaning that the training sample did not include any stable glaciers with those characteristics, but only pulsating ones. Conversely, small values of $K$ and $C$ are associated with lower probabilities of pulsation; however, increases of $C$ above 4 km appear to again suppress the probability



to pulsate. Calculating probabilities of pulsation with also the third predictor ($I_T$, the mean slope of the tongue) yields very similar results to those of the simpler model (Fig. A1e), indicating low sensitivity to that parameter – hence a minor influence over pulsating behavior. Specifically, the effect of slope appears to be a generalized increase of the probability of pulsation with increasing slope, particularly marked for large values of $K$ (Fig. A1f/g).

**Appendix B: Lists of satellite scenes**

Here, we provide the list (Table B1) of declassified scenes from reconnaissance satellites that we used in the study.

**Table B1.** Declassified scenes from reconnaissance satellites used in our study, listed by chronological order of acquisition.

| Date | Sensor | Identifier |
|------|--------|------------|
| 18/08/1968 | KH-4 PC | DS1104-2169DF090–093 |
| | | DS1104-2169DA096–099 |
| 15/09/1971 | KH-4 PC | DS1115-1072DF185 |
| | | DS1115-1072DA191 |
| 16/07/1973 | KH-9 MC | DZB1206-500007L015001 |
| | | DZB1206-500007L016001 |
| 03/08/1973 | KH-9 MC | DZB1206-500080L016001 |
| | | DZB1206-500080L017001 |
| | | DZB1206-500080L018001 |
| | | DZB1206-500080L019001 |
| 22/11/1973 | KH-9 MC | DZB1207-500041L001001 |
| | | DZB1207-500041L002001 |
| | | DZB1207-500041L003001 |
| 25/07/1980 | KH-9 PC | D3C1216-200279F036–039 |
| | | D3C1216-200279A037–040 |
| 20/08/1980 | KH-9 MC | DZB1216-500273L006001 |
| | | DZB1216-500273L007001 |
| | | DZB1216-500273L008001 |

**Appendix C: Metrics of classifier performance**

Here we provide the formal definitions of the performance metrics used to evaluate the GS1980 classifier. Variables appearing in the formulas are defined as in Sect. 4.5.



Recall, also known as sensitivity or True Positive Rate (TPR):

$$TPR = \frac{TP}{TP + FN} \qquad (C1)$$

Specificity, also known as True Negative Rate (TNR):

$$TNR = \frac{TN}{TN + FP} \qquad (C2)$$

False Positive Rate (FPR):

$$FPR = \frac{FP}{FP + TN} \qquad (C3)$$

Accuracy:

$$Acc = \frac{TP + TN}{TP + TN + FP + FN} \qquad (C4)$$

Balanced Accuracy:

$$BA = \frac{1}{2}\left(\frac{TP}{TP + FN} + \frac{TN}{TN + FP}\right) \qquad (C5)$$

Precision, also known as Positive Predictive Value (PPV):

$$PPV = \frac{TP}{TP + FP} \qquad (C6)$$

F1 score:

$$F_1 = 2 \times \frac{PPV \times TPR}{PPV + TPR} \qquad (C7)$$

*Author contributions.* EM designed the study, compiled the datasets, performed the analysis, and led the writing with contributions from all
authors. AB and JK processed the KH-4 data, SG processed the KH-9 PC data. MB and MH provided regional expertise and data of previous
studies at Abramov glacier. All authors contributed to the interpretation of the results.

*Competing interests.* The authors declare that they have no conflict of interest.

*Acknowledgements.* We thank the project "Strengthening the resilience of Central Asian countries by enabling regional cooperation to
assess glacio-nival systems to develop integrated methods for sustainable development and adaptation to climate change" funded by the
Global Environment Facility / United Nations Development Programme / United Nations Educational, Scientific and cultural Organization
(GEF/UNDP/UNESCO, contract no. 4500484501) and the project "Cryospheric Observation and Modelling for Improved Adaptation in
Central Asia (CROMO-ADAPT)" (contract no. 81072443) funded by the Swiss Agency for Development and Cooperation and the University



of Fribourg and the project "From ice to microorganisms and humans: Toward an interdisciplinary understanding of climate change impacts on the Third Pole (PAMIR)" (grant number: SPI-FLAG-2021-001) funded by the SPI Flagship Initiative of the Swiss Polar Institute. We

would like to thank Amaury Dehecq for the processed KH-9 MC data, Christine Bichsel for the instructive discussions about Soviet research in the Hissar-Alay, and Gulomjon Umirzakov for access to Russian-language literature. We acknowledge the use of SPOT images acquired by CNES's Spot World Heritage Programme, as well as the Education and Research Program of Planet Labs for access to the RapidEye scenes. AB acknowledges the research funding (grant no. CRG/2021/002450) received from Science & Engineering Research Board (SERB), now Anusandhan National Research Foundation (ANRF), Department of Science & Technology (DST), India, under Core Research Grant (CRG).





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





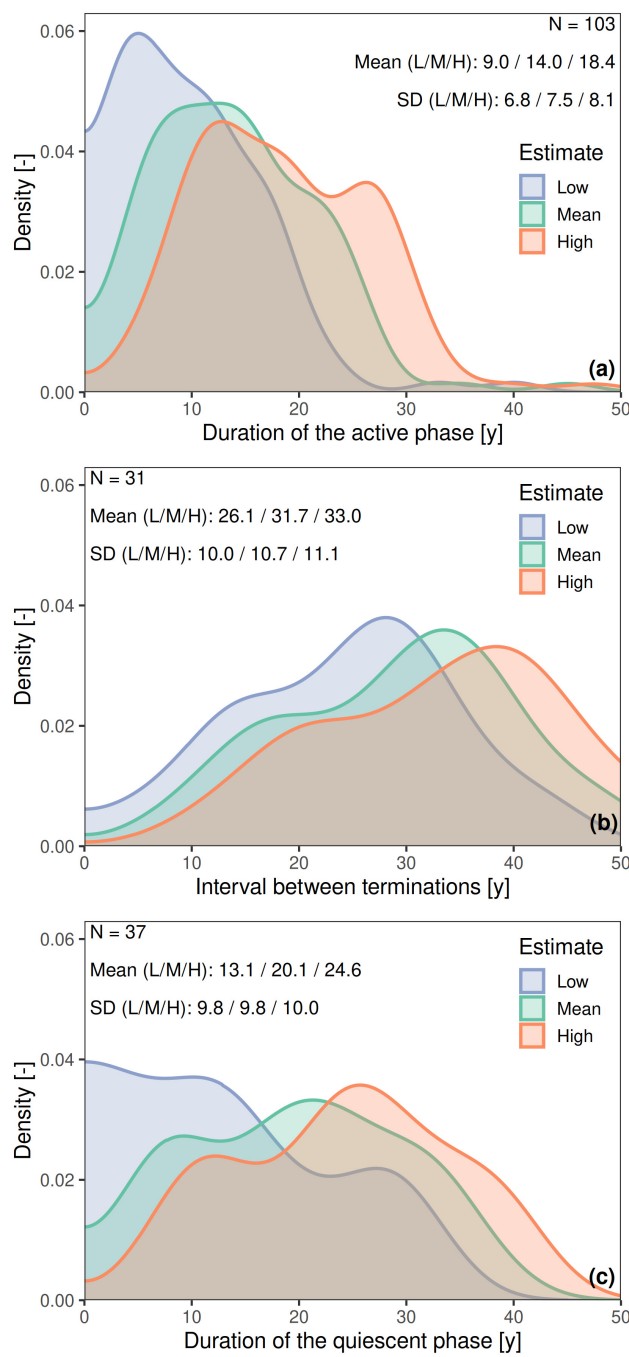

**Figure 7. (a)** Distribution of durations of the active phase, only for pulsations fully contained within our dataset. **(b)** Distribution of intervals between termination of consecutive pulsations, only for glaciers with two or three observed pulsations. **(c)** Distribution of durations of the quiescent phase, only for glaciers with fully observed quiescent phases between two consecutive pulsations. See Sect. 4.6 for the definition of low/mean/high (L/M/H) estimates.



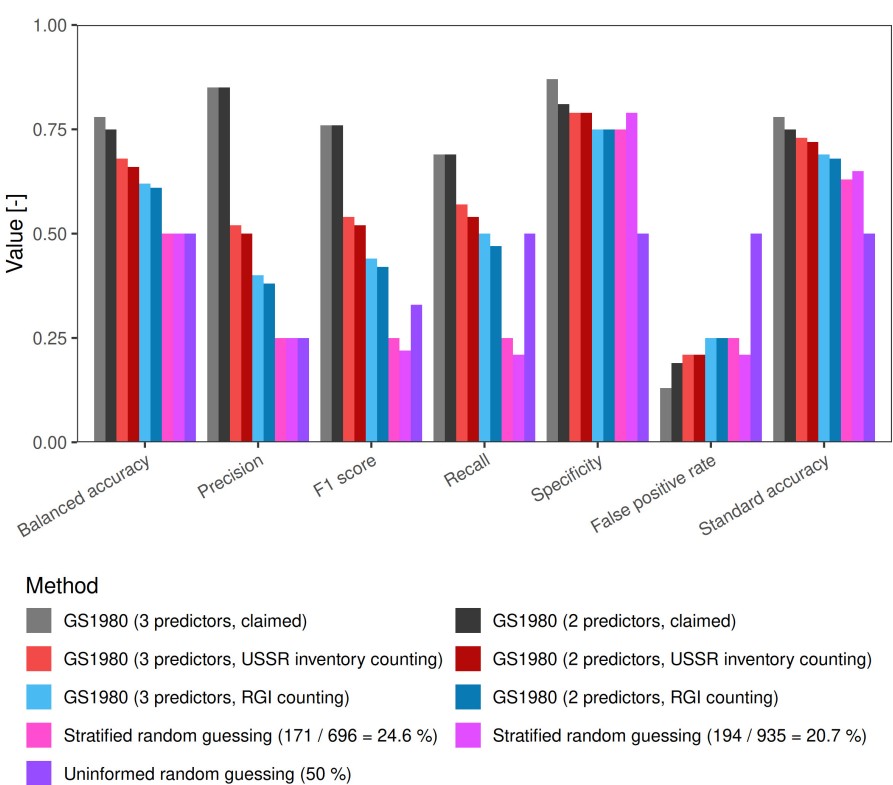

**Figure 8.** Performance metrics of the GS1980 classification method compared to different versions of random guessing. The first three metrics account specifically for class imbalance. The "claimed" performance refers to the leave-one-out validation results reported in Glazirin' and Shchetinnikov (1980). See Sect. 4.1 for the description of counting methods (USSR inventory and RGI). The two fractions of pulsating glaciers selected for stratified random guessing correspond to the result found in our study and to the GS1980 prediction, respectively.



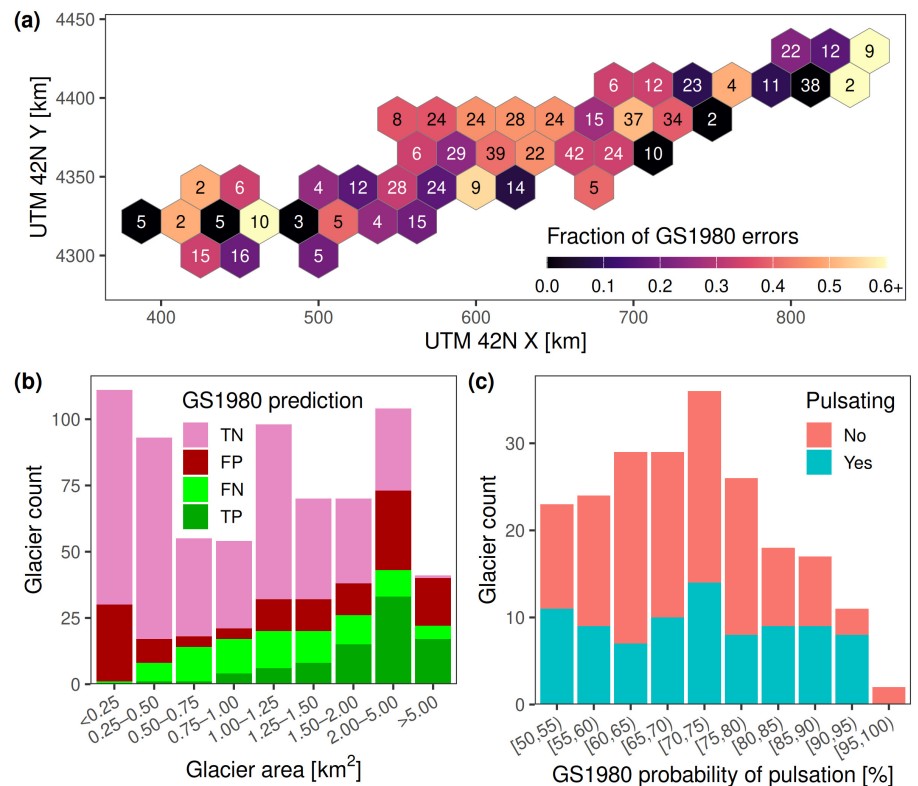

**Figure 9. (a)** Fraction of GS1980 misclassifications (false positives and false negatives) within the examined sample over the Hissar-Alay. The number of examined glaciers is inscribed in each tile. **(b)** Breakdown of GS1980 performance by glacier area class. TN: true negatives. FP: false positives. FN: false negatives. TP: true positives. Also see Fig. 5. **(c)** Fraction of pulsating and stable glaciers by predicted probability of pulsation of GS1980. The three plots depict the results of the GS1980 three-predictors method (Sect. 3.1), with RGI7.0 aggregation (Sect. 4.1).

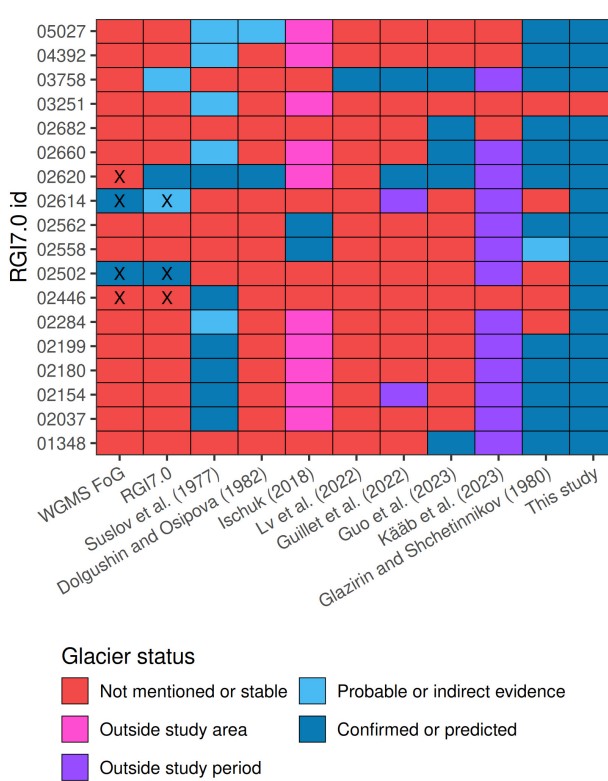

**Figure 10.** Comprehensive list of glacier instabilities reported over the Hissar-Alay within previous studies. Indexing mismatches found by our analysis are marked with an X. "Outside study period" indicates glaciers where we observed a pulsation taking place outside the period included in the respective study. FoG is the Fluctuations of Glaciers product (World Glacier Monitoring Service, 2024). Glacier 03251 was found to be stable within our analysis, but terminating in a large rock glacier whose active geomorphology may have led to its misclassification by Suslov et al. (1977). All indices were shortened by removing the full prefix "RGI2000-v7.0-G-13-".



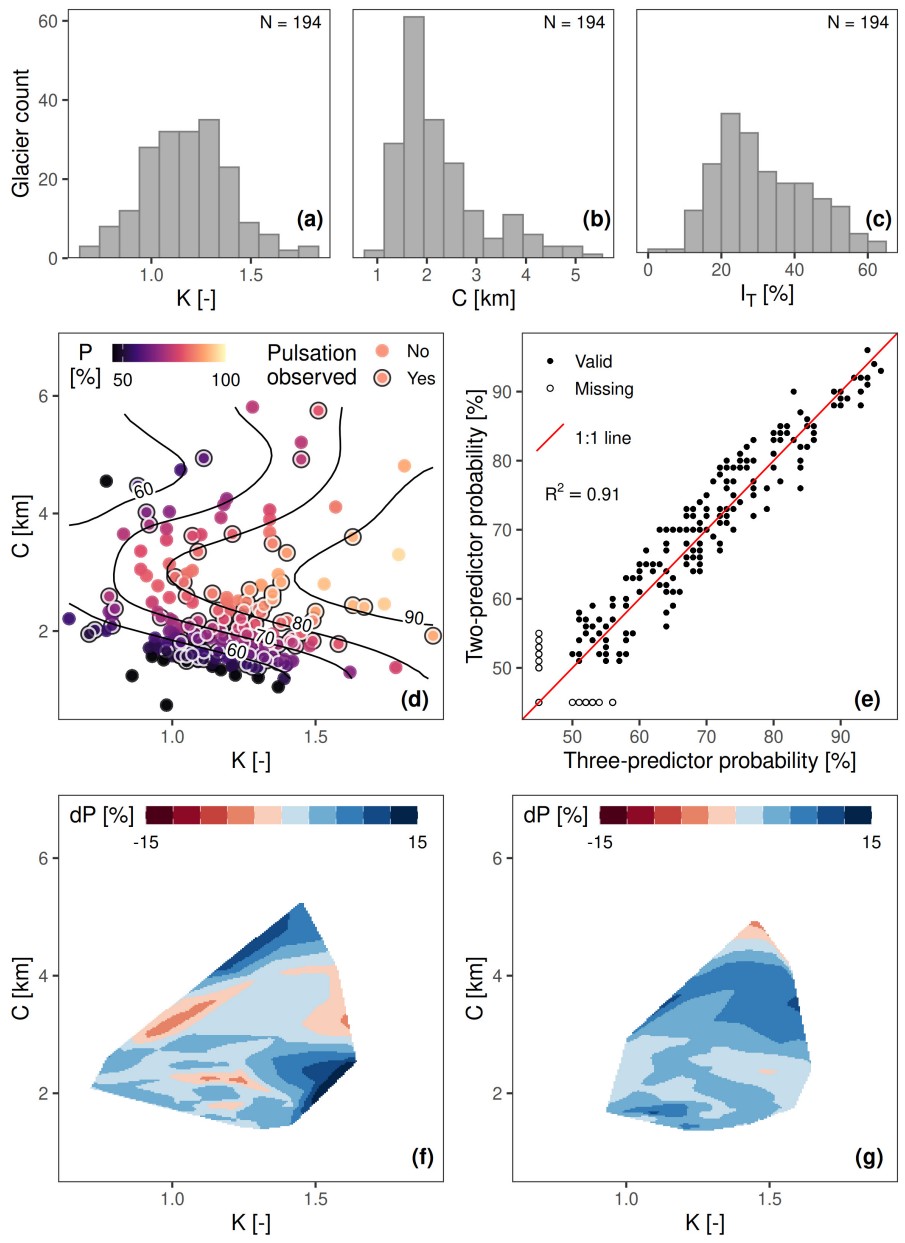

**Figure A1.** Overview of the GS1980 classifier of glacier pulsation in the Hissar-Alay. **(a)**, **(b)**, **(c)** Distribution of morphological parameters $K$ (ratio of accumulation to ablation area extents), $C$ (ratio of accumulation area extent to tongue width), and $I_T$ (mean slope of the tongue), over all glaciers predicted to be pulsating by GS1980. The values are directly extracted from the table of GS1980. **(d)** Calculated probabilities of pulsation with the two-predictor set $(K, C)$. Pulsating glaciers observed within our study are highlighted. **(e)** Scatterplot of calculated probabilities with the two sets of predictors: $(K, C, I_T)$, and $(K, C)$, respectively. "Missing" indicates GS1980 entries where the calculated probability is below 50 % (and thus not reported) with one set of predictors: those instances are assigned a value of 45 % for display purposes. **(f)**, **(g)** Slices (in predictor space) showing the dependence of the probability of pulsation on the third parameter ($I_T$, mean slope of the tongue). Colors represent the change of probability calculated over tertiles of increasing surface slope: **(f)** change between the first and second tertile, and **(g)** change between the second and third tertile. Probabilities are interpolated bilinearly over the convex hull of all points belonging to the corresponding slope tertiles.