# Peer review of "Predicted and observed glacier pulsations in the Hissar-Alay of Central Asia"

_EGUsphere, 2025_

## Referee Comment (RC1)

**Review for**

**Predicted and observed glacier pulsations in the Hissar-Alay of Central Asia Mattea and others, 2025**

The authors revisit the documentation of glacier flow instabilities in Central Asia, specifically the Hissar-Alay range, by reassessing the predictions of classes of pulsating behavior made by Glazirin and Schetinnikov (1980). They find that the model predictions are less accurate than previously reported due to the assumptions of the model and uncertainty in the required glacier morphological data (AAR, glacier tongue surface slope, etc.).

Based on the author's findings, the model itself has significant limitations and does not appear to be worthwhile to implement in other contexts. The expansive remote sensing records used to thoroughly document pulsation in the Hissar-Alay and findings on the dominant pulsation behavior in the region (slow, small advances and long active phases) do seem to be valuable. The Data Availability section does not make it clear whether or not these glacier dynamics datasets will be archived in any public repositories. There are few known regional-scale maps of thickness change and time series of glacier length change – those, and even the author's updated regional glacier inventory, would be a significant contribution to the glaciological community. I suggest reducing the emphasis on the modeling in the manuscript and insist that the authors archive their datasets. Reframing the manuscript, to focus on the content in the second half of the abstract (L9-L17) for example, and archiving the datasets may take some time. Therefore, I suggest a major revision.

The writing itself is quite clear and easy to read and the figures are good. Below are handful of specific comments on the manuscript as it stands currently:

- L30: Suggest adding citations of Beaud et al., 2021 (<a href="https://doi.org/10.5194/tc-16-3123-2022">https://doi.org/10.1017/jog.2023.99</a>)
- L178/9: Add "due to its small size" or something similar
- L376-382: Clarify what you mean by interval here, surge interval?
- Fig. 1. Make the Abramov glacier region inset much larger so that it's easier to see the color classifications.